



# Brief communication: A simple axial induction modification to WRF's Fitch wind farm parameterisation

Lukas Vollmer[1,*], Balthazar Arnoldus Maria Sengers[1,*], and Martin Dörenkämper[1]

[1]Fraunhofer IWES, Küpkersweg 70, 26129 Oldenburg, Germany
[*]These authors contributed equally to this work.

**Correspondence:** Lukas Vollmer (lukas.vollmer@iwes.fraunhofer.de); Balthazar Sengers
(balthazar.sengers@iwes.fraunhofer.de)

**Abstract.** We propose a modification to the Fitch wind farm parameterisation implemented in the Weather Research and Forecasting (WRF) model. This modification, derived from the 1D momentum theory, employs a wind speed dependent induction factor to correct the local grid wind speed back to free stream, before computing the turbine's power and thrust. While the original implementation underestimates power, the modified version shows a better agreement with the power curve. We strongly
recommend the modification to be employed for all studies that model not more than one turbine per WRF grid cell.

## 1 Introduction

As offshore wind energy is developing quickly and in relatively concentrated regions along the coastline, models are needed that correctly represent large-scale wake effects and their interactions with the atmosphere. Several attempts towards the realistic modelling of those effects have been made by employing fast engineering (e.g., Nygaard et al., 2020), high-fidelity
(e.g., Wiegant and Verzijlbergh, 2019; Maas and Raasch, 2022) or mesoscale weather models (e.g., Lundquist et al., 2019; Siedersleben et al., 2018). The latter are being used by an increasing number of institutions (Fischereit et al., 2021).

The most commonly used mesoscale model, especially when studying large-scale wake effects, is the Weather Research and Forecasting model (WRF, Skamarock et al. (2021)). The wind farm parameterisation proposed by Fitch et al. (2012) has been integrated into WRF's main code for approximately a decade, establishing itself as the most frequently utilized approach
(Fischereit et al., 2021). It models wind farms as an elevated sink of momentum and a source of turbulent kinetic energy, assuming that thrust that is not converted to power linearly scales with the turbulence added to the flow.

The Fitch parameterisation does not consider the effect induction has on the local wind speed at the grid cell of the turbine. In this publication we show that, as a consequence, the turbines' power and thrust are underestimated. Abkar and Porté-Agel (2015) mention that the local wind speed may deviate from the free wind speed that should be used for the calculation of
forces and power production. However, they only discuss this effect for cases of multiple wind turbines in a grid cell and use high-fidelity simulations to compute a correction factor. A similar approach was suggested by Mayol et al. (2020), who introduced an induction-aware modification to the Fitch parameterisation by computing a correction factor with idealized WRF simulations.





The effect of axial induction increases in relevance with growing ratios between turbine dimensions and grid sizes. It is therefore
desirable to have a method that does not rely on precomputed correction factors, but rather one that is directly generalizable.
This brief communication proposes a physics-derived modification based on 1D momentum theory. It considers the induction
factor of the wind turbine to correct the local grid wind speed back to a free stream wind speed, as is standard in actuator disc
modelling. Verification is done by comparing the model's power estimations to power calculations using the wind speed from
a reference simulation without a turbine.

## 2  Methodology

### 2.1  WRF setup

WRF version 4.2.1 was employed for this study. Note that this version already includes the bug fix to the Fitch parameterisation
reported by Archer et al. (2020). The simulation setup was largely based on Cañadillas et al. (2022). Three one-way nested
domains, the smallest one having a spatial resolution of 2 km, were centered around a random point in the German Bight.
Initial and boundary conditions were described every 6 hours by ERA5, while the sea surface temperature was provided by
OSTIA. The physics schemes used consists of the following: the MYNN 2.5 level planetary boundary layer scheme, the Noah
Land-Surface model, the MYNN surface layer scheme, the RRTMG long-wave and short-wave radiation schemes, the WRF
Single-Moment 5-class microphysics scheme and the Kain-Fritsch cumulus scheme (only outer two domains).

We conducted a simulation of a single day (February 15th, 2020), of which the first 6 hours were omitted as spin-up, after
which the following 12 hours were used for analysis. This day was chosen as wind speeds consistently decreased throughout,
allowing for a reconstruction of the power curve. A single turbine was placed in the domain, which was centered around the
German Bight. To analyse the sensitivity of the power calculations on the turbine dimensions, two turbine types were used: the
NREL 5 MW wind turbine (Jonkman et al., 2009) with a hub height of 90 m and a rotor diameter of 126 m and a 22 MW wind
turbine with a hub height of 175 m and a rotor diameter of 290 m that was created by simple dimensional upscaling of the IEA
15 MW wind turbine (Evan Gaertner et al., 2020). As a reference, a simulation without turbines was performed, from which
wind speed time series at hub height were extracted for a direct calculation of power by means of the turbine's power curve.

### 2.2  Fitch modification

The Fitch parameterisation calculates each turbine's power output $P$ and its influence on the momentum equations using the
following equations:

$$\frac{\partial u}{\partial t} = -0.5 \cdot C_{\mathrm{T}}(u) \cdot u^2 \cdot A \tag{1}$$

$$P = 0.5 \cdot \rho \cdot C_{\mathrm{P}}(u) \cdot u^3 \cdot A \tag{2}$$

$$\frac{\partial \mathrm{TKE}}{\partial t} = -0.5 \cdot C_{\mathrm{TKE}}(u) \cdot u^3 \cdot A \tag{3}$$

with $u$ the locally sampled wind speed in the grid cell in which the turbine is located, $C_{\mathrm{T}}$ and $C_{\mathrm{P}}$ the turbine's wind speed-
dependent thrust and power coefficients, $A = \pi(D/2)^2$ the turbine's rotor area and $\rho$ a standard air density. The coefficient





$C_{\mathrm{TKE}}$, used to calculate the tendency of the turbulent kinetic energy (TKE), is derived from the known power and thrust coefficients by $C_{\mathrm{TKE}} = C_{\mathrm{T}} - C_{\mathrm{P}}$. These calculations are done for each turbine individually. When one grid cell contains multiple turbines, the resulting effect on the momentum equations is just a sum of the single turbine contributions.

$C_{\mathrm{T}}$ and $C_{\mathrm{P}}$ are dependent on a free stream wind speed that is undisturbed by the presence of the turbine. In practice, measurements are taken at least 2.5 rotor diameter upstream of the wind turbine. As the current implementation of the Fitch equations

samples the wind speed $u$ inside the grid cell, this condition is not respected. To this end, we suggest a modification to the Fitch equations that is based on a correction of the local wind speed $u$ of the grid cell in which the turbine is placed, by the induction factor $a$ of the wind turbine. The aim is to obtain a free wind speed $u_{\infty}$ to be used in power and thrust calculations:

$$u_{\infty} = \frac{u}{1-a} \qquad (4)$$

with

$$a = 0.5(1 - \sqrt{1 - C_{\mathrm{T}}(u_{\infty})}) \cdot f(\delta, \mathrm{d}x, D) \qquad (5)$$

The induction factor $a$ is calculated from the $C_{\mathrm{T}}$ of the turbine. A correction function $f$ is needed to calculate how much of the mesh cross-section is occupied by the turbine area $A$. Because turbine orientation and mesh orientation rarely align, we propose the following correction function, which considers the local wind direction $\delta$ at hub height:

$$f(u, \mathrm{d}x, D) = A \cdot (D \cdot \mathrm{d}x \cdot \min(|\frac{1}{\cos(\delta)}|, |\frac{1}{\sin(\delta)}|))^{-1} \qquad (6)$$

with $\mathrm{d}x$ the horizontal grid size, $D$ the rotor diameter and $\delta$ the horizontal wind direction. The correction function $f$ becomes $A \cdot (D \cdot \mathrm{d}x)^{-1}$ when the wind vector and thus the turbine orientation is perpendicular to the mesh and $A \cdot (D \cdot \sqrt{2} \cdot \mathrm{d}x)^{-1}$ when it is diagonal to the mesh. In the final set of equations, the wind speed variable $u_{\infty}$ replaces the locally sampled wind speed $u$ in the equations 1 to 3.

For $n$ wind turbines within one grid cell, the wind speed correction can be extended by multiplying each turbine's induction.

Note that we consider here that each turbine in the grid cell faces the free wind speed, and no mutual wake interactions occur. The assumption also is only valid when all wind turbines are of the same type.

$$u_{\infty} = \frac{u}{(1-a)^n} \qquad (7)$$

## 3 Results

Figure 1(a,d) exhibits the time series of power production for the reference calculations, the original Fitch (Fitch-o), and the

induction modification proposed here (Fitch-AIF). Although the time series do not align perfectly due to the divergence of the calculations related to the different boundary conditions (the inclusion of the turbine), they reveal that Fitch-o clearly underestimates the generated power while Fitch-AIF more accurately matches the reference. This finding is strengthened by the reconstructed power curves displayed in Fig. 1(b,e), which were calculated by using the undisturbed wind speeds of the adjacent upstream grid cell.

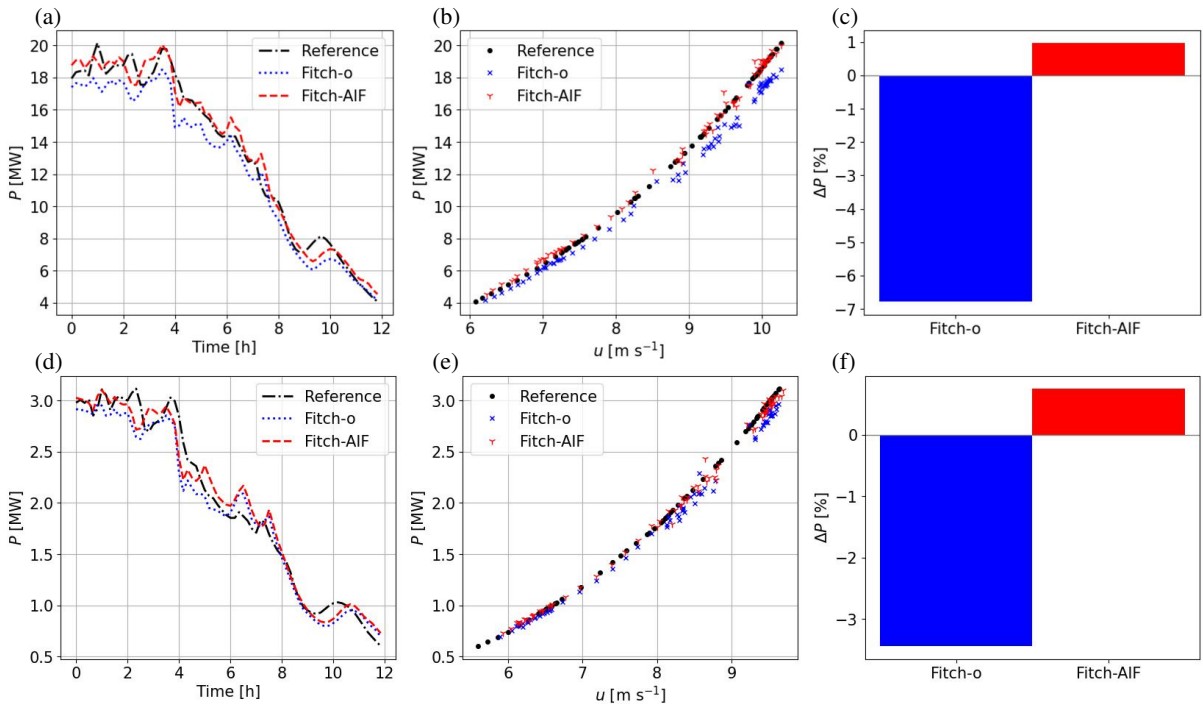

**Figure 1.** Comparison for the 22 MW (a-c) and 5 MW (d-f) turbines. (a,d) Power time series of the modelled wind turbines from different versions of the Fitch model and the reference. (b,e) Reconstructed power curves. (c,f) Mean differences between the two model versions to the reference power calculation.

Figures 1(c,f) compare the mean power over the 12 h long simulation relative to the reference. As also clear from the other two visualisations in Fig. 1, the power difference is lower for the smaller turbine, which can be explained by the lower induction effect generated by the smaller turbine. Compared to Fitch-o, Fitch-AIF shows a power and thrust increase of 8.3% and 5.4% respectively for the 22 MW turbine and 4.3% and 2.7% respectively for the 5 MW turbine.

To validate the modification for multiple turbines, five 22 MW wind turbines were placed within a single grid cell in WRF. Under the assumption that all five turbines operate in free wind conditions, the average power production per turbine calculated with the modification for multiple turbines (Fitch-mAIF) was compared with Fitch-o, Fitch-AIF and the reference (Fig. 2). This exercise reveals that the underestimation of Fitch-o for this extreme dense case of turbines is in the order of 25 %. This reduces to about 20 % for Fitch-AIF but is almost eliminated when accounting for the number of turbines in the grid cell (Fitch-mAIF). Compared to Fitch-o, Fitch-AIF shows a power and thrust increase of 7.1% and 4.5%, while Fitch-mAIF displays increases of and 39.6% and 24.3%. Although this case might not be very realistic, as the turbines physically barely fit into the grid cell, it demonstrates the scalability of the proposed modification.





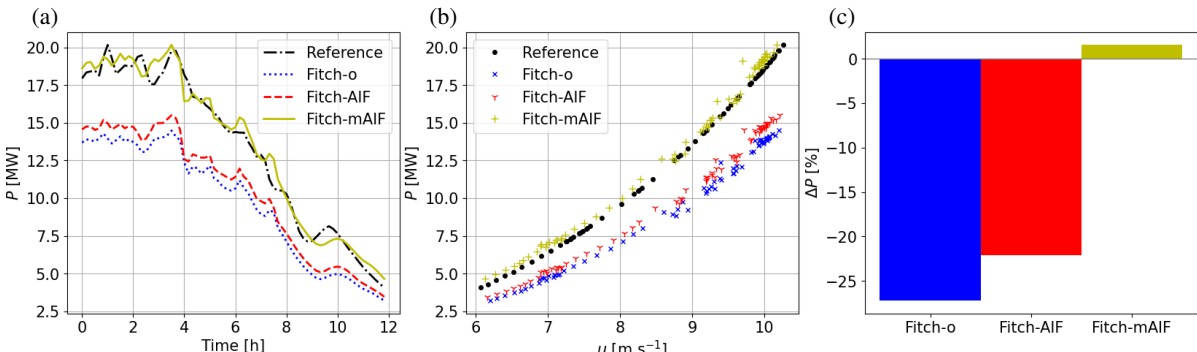

**Figure 2.** Same as in Fig.1 but for five 22 MW turbines per grid cell and with the addition of the Fitch-mAIF modification that combines the induction effect of the turbines within the grid cell for power calculation.

## 4 Conclusions

The Fitch wind farm parameterisation implemented in WRF version 4.2.1 does not account for local induction effects and consequently underestimates power production of a single wind turbine in the dynamic region of the power curve. This issue

is amplified for large turbines or when there are multiple turbines in one grid cell. To correct this underestimation, we propose a modification derived from the 1D momentum theory. Instead of using the local wind speed of the grid cell, we use the wind speed dependent induction factor to estimate the free stream wind speed. Results from a simple analysis show that the turbine's power curve is better reproduced when including this modification.

It is important to note that downstream wind speeds from wind farms modelled with WRF-Fitch have shown a good agreement

with measurements (Cañadillas et al., 2022; Fischereit et al., 2021). This implies that the Fitch model may be unintentionally generating correct results by ignoring induction as well as wake effects for grid cells containing multiple turbines. We suggest that the model modification should be thoroughly evaluated in these cases. However, in scenarios with a single turbine per cell (e.g., calculations involving future turbine dimensions), we strongly recommend using the proposed modification for more accurate yield and wind resource assessments.

*Code availability.* The WRF code with the induction correction for the Fitch parameterisation will be made available when pending legal requirements are resolved.

*Author contributions.* LV conceptualized the idea, BAMS implemented of the correction in WRF and performed the simulations. MD initiated the associated research project, and was thus involved in the funding acquisition as well as discussions. All authors contributed intensively to the writing and reviewing of the manuscript.



*Competing interests.*  The authors declare that there are no competing interests

*Acknowledgements.*  The results presented in this paper were derived in the framework of the X-Wakes (grant no. 03EE3008) project. The X-Wakes project is funded by the German Federal Ministry for Economic Affairs and Climate Action (Bundesministerium für Wirtschaft und Klimaschutz – BMWK) due to a decision of the German Bundestag. The simulations were partly performed at the HPC Cluster EDDY, located at the University of Oldenburg (Germany) funded by BMWK (grant no. 03240).





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
