# Peer review of "Brief communication: A simple axial induction modification to WRF's Fitch wind farm parameterisation"

_Wind Energy Science, 2023_

## Author Comment (AC1)

**Response to Reviewer 1:**

Dear Patrick,

thanks for your review of our manuscript. You find our answers to your comments (black) in the following in red.

The authors address a very important shortcoming of the highly-used wind farm parameterization within the WRF model. It has been fairly well known that the Fitch wind farm parameterization underestimates power. While other studies have attempted to overcome this issue by using correction factors, this study presents a generalizable fix by using the free-stream wind speed which should be very beneficial to the modeling and resource assessment community. I have a few comments of which I believe would be beneficial for the paper, but are not critical for publication. That said, I recommend the article for publication.

**Specific comments**

- Many of the uses for the wind farm parameterization in the wind energy industry involve the impact of wake effects on other wind farms or prospective lease areas. The final paragraph mentions how well the Fitch scheme does with wakes over land but does not mention how the new fix impacts wake resolution. Given that there are no observations for comparison, what does the difference in mean wind speed (over, say, the 12 hour analysis) look like between the Fitch-o, -AIF, and -mAIF cases? Are they drastically different or still quite similar? It will not be a quantitative result, but might shed light on if the wake impact will be large or small.
  In the manuscript we mention that not only power but also thrust of the turbines increases (ll. 87/88 & ll. 94/95). The increase of thrust of course also leads to increased wake deficits downstream of the turbines. We added a sentence to make this clear. The wake deficit scales with the increase in thrust. A quantification based on the now-used five-day simulation (see next comment) is not meaningful as the wind direction changes over time and thus the grid cells that are influenced by the wake also change. Quantifying the wake deficit averaged over a longer period of time with the same wind direction would call for a much longer simulation and is considered out of the scope of the current work.

- The change to the Fitch parameterization now over-predicts power. It is much closer and clearly beneficial, but I am curious as to if the authors can explain why it is over-predicting power.
  You are correct that the results now show that the model slightly overpredicts power. This is due to the 'butterfly effect' in WRF. The reference simulation and the simulations with the turbines differ because the presence of the turbine introduces a disturbance that results in a slight phase shift between the simulations. To statistically average this effect, we performed longer simulations of 5 days for the revised manuscript. The results of these simulations show that the difference between the Fitch-AIF and the reference becomes negligible (see updated figures).

- This is a very interesting and potentially beneficial update to the Fitch parameterization. A short section of shortcomings and future work would be nice to add.
  Thanks, as suggested we added more discussion in the last section of the revised manuscript.

**Technical Corrections**

- Line 95 "of and" should be "of"
  Fixed

**Response to Reviewer 2:**

Dear Referee #2,

thank you for the review of our manuscript. Please find our answers to your comments (black) below in red.

Review of the manuscript: "Brief communication: A simple axial induction modification to WRF's Fitch wind farm parameterization" by Lukas Vollmer, Balthazar Arnoldus Maria Sengers, and Martin Dörenkämper submitted for publication in Wind Energy Science.

In the manuscript "Brief communication: A simple axial induction modification to WRF's Fitch wind farm parameterization" the authors present a modification to the Fitch wind plant parameterization implemented in the Weather Research and Forecasting model to account for the decrease in wind speed in the presence of a wind turbine compared to the free stream wind speed. The authors propose, implement, and demonstrate the effect of induction factor parameterization.

**General Remarks**

Wind plant parameterization play an important role accurate resource assessment and prediction of power production. The study presented in the manuscript therefore addresses a potentially important topic. The argument for a need to account for wind speed reduction in a presence of a wind turbine compared to the free stream wind speed and therefore for modifying a widely used parameterization are well justified. However, the presented study is lacking in a detailed analysis. Furthermore, it does not address some unexpected results, e.g., that a wind plant parameterization with an induction factor results in higher power output compared to the reference simulation which does not include a wind turbine. Finally, it includes an unrealistic simulation with five 22 MW wind turbines in a 2 km x 2 km grid cell results of which do not provide any useful insight. It is not clear why would one use such a case when there are available data from operating wind farms (e.g., Ali et al. 2023, https://doi-org.cuucar.idm.oclc.org/10.1175/MWR-D-23-0006.1) which can be used to compare wind plant parameterizations. Additional comments are provided blow under Specific Remarks.

Generally, you write that the manuscript lacks a validation with measurement data from operating wind farm and request that the study should be redesigned to include this data. While we agree that validation with wind farm data is an important aspect in the development of models for resource assessment, we believe that doing so within the context of the presented brief communication paper would not be appropriate.
The paper describes an underlying short-coming in the model physics of the Fitch wind farm parametrisation used in WRF, and increasingly also in other numerical weather models. The validity of the proposed modification can directly be verified with sanity or plausibility checks, as done in the manuscript, and does not require a validation with measurements. The included sanity check clearly demonstrates that the wind turbine's power production is not correctly reproduced with the default Fitch model, while the suggested correction increases accuracy significantly. We deliberately chose the brief-communication format to focus on the importance of this lack of model physics in the original Fitch parametrisation.

Besides, we argue that a validation with measurement data is not the appropriate method for the objective of this manuscript, as this would introduce large uncertainties related to the site-dependent power curve, the modelling of the wind speed with the numerical model, the data processing etc. We believe that this would water down the conclusions drawn in the paper.
We acknowledge that the scope of the paper is apparently not well enough described and clarified this in the revised manuscript.

With the example of 5 22 MW turbines in one cell we aim to demonstrate that the model correction scales very well also with high power densities and multiple turbines per grid cell. We deliberately selected a rather unrealistic scenario to show that the correction is plausible also in extreme cases. We also reformulated the intention of this comparison for clarity.

Taking all the above into account in addition I do not recommend the manuscript for publication. However, I would encourage the authors to resubmit the manuscript after they redesign the study and include comparison with observed data in addition to providing a more detailed analysis, followed by potential refinement of the axial induction addressing over-prediction of power.

**Specific Remarks**

- Line 5 – This statement can be confusing; it would be better to reword it as follows: "... model only one turbine per WRF gird cell." However, the question is why is this not recommended if there are more turbines per grid cell?
  we added a sentence here that mentions that for simulations with more than one turbine per grid cell, inner-cell wake losses have to be considered.
- Line 13 – It would be important to reference the original journal publication Skamarock and Klemp 2008, doi:10.1016/j.jcp.2007.01.037.
  Thanks for the suggestion, we added the reference.
- Line 24 – Instead of "growing" better would be "increasing."
  Changed
- Line 38 – It would be important to provide information about the surface layer, i.e. surface roughness parameterization used over the ocean?
  The manuscript detailed that we used the MYNN surface layer scheme, which uses the Charnock relation for surface roughness over water. We used the default settings therein and clarifications have therefore not been added to the manuscript.
- Line 67 – However, A/(D*dx) is not that ratio - it should be A/(dx*dz). Either the name of the ratio should be modified, or its definition.
  We correct the induction factor by the ratio of rotor area to the cross-section of the model on which the thrust is applied. For flow perpendicular to one grid cell, it would look like the sketch below. In the end it is an approximation and no name was specifically give to this ratio. We think the way we formulate it makes the most sense. We did not include this sketch in the revised

manuscript but this could be done upon request.

[Figure]

- Figure 1 – The panels (b, e) are not very clear, it would be better to plot power difference w.r.t. reference vs. wind speed. Also, it seems that Fitch-AIF frequently results in more power than reference, how can that be explained or justified? The panels (c, f) are not very informative, the same information can be conveyed more succinctly with one simple sentence. However, it is not clear why is the power production higher with Fitch-AIF then reference.
  Thanks for the suggestion, we revised this part and selected different visualisations.
  The power overprediction by Fitch-AIF in the 12-h time series is due to the 'butterfly effect' in WRF. The reference simulation and the simulations with the turbines differ because the presence of the turbine introduces a disturbance that results in a slight phase shift between the simulations. To statistically average this effect, we executed a longer simulation of 5 days for the revised paper. In the results of this simulations the difference between the Fitch-AIF and the reference becomes negligible.
- Line 94 – It is not clear what is meant by "accounting", is the same "accounting" applied to Fitch-o?
  we don't really understand this comment. We now wrote "consider".
- Line 96 – five 22 MW turbines in 2 km x 2 km, is beyond unrealistic. So why was this case selected if it is "not very realistic?"
  we rewrote this part as explained earlier

---

## Author Response (AR2)

**Response to Reviewer 2, round 2:**

Dear Referee #2,

thank you for your review of our manuscript in this second round. Please find our answers to your comments (black) below in red and changes made to the manuscript in blue.

In the revised manuscript the authors made significant modification to the figures and the text, in particular the presentation of results and the conclusion. In general, these modifications resulted in improvement of the manuscript. The authors adequately addressed all my minor comments; however, they dismissed the major comments related to the overprediction of power because of the "butterfly effect" after performing a longer-term simulation and justified using unreasonable wind turbine density as demonstrating scaling of the implemented induction modification. However, as can be seen in the Figure 2 b) on average the simulation with the induction modification results in larger power than the reference simulation, for the range of wind speeds between 4 m/s and approximately 11 m/s. It is not clear how can this be attributed to the "butterfly effect," i.e. dependence on initial conditions, especially since the result is based on five-day simulation, and time average should converge to an ensemble average. Furthermore, because of the over prediction, this unrealistic simulation with 22 MW turbines does not really demonstrate scalability. The question is what is the cause of this over prediction? The authors should provide a more convincing explanation before the manuscript can be accepted for publication.

We appreciate the critical evaluation of this part of the manuscript. It becomes clear to us that while such artifacts are well-known in the WRF community, it does make sense to add clarifications for the reader of the manuscript.
The presence of the turbines initially disturbs the flow only locally, but when solving for the momentum equations this local disturbance also affects the flow in the rest of the domain, also upstream of the turbines which further in the simulation then again affects the wind speed experienced by the turbines. This results in a decorrelation of the wind speed between the simulations with and without turbines. Even though this effect is small, it is noticeable in short simulations, like the 12h simulations we showed in the first iteration of the manuscript. For longer simulations this effect will vanish. We aimed to demonstrate this with a 5-day long simulation in the second iteration of the manuscript, but evidently this is still too short to fully eliminate this effect, illustrated by the small overestimation of power compared to the reference in Figure 2c. Not that this difference is only 0.32 % or 38.3 W per turbine, which even at the shallowest part of region 2 of the power curve could be caused by a wind speed difference as small as 0.056 m/s.
Focusing on Figure 2b specifically, as referenced by the reviewer, it should be reiterated that the wind speed used here on the x-axis is derived from the reference simulation without turbines. Between 5 m/s and 11 m/s both positive and negative $\Delta P$ can be found that cancel each other out, with values in the order of 10% caused by the decorrelation of wind speed between the two separate simulations. More intriguing is the apparent systematic positive bias between 3 and 5 m/s, which can be explained as follows. A deviation (due to decorrelation) of the wind speed in the simulation with turbines from the reference simulation without turbines results in large differences in $\Delta P$. This can be because the wind speed drops below cut-in (resulting in $\Delta P$ = -100%, not shown in this plot) or because the reference

power is very small (resulting in large positive values of $\Delta$P because the value in the denominator is small). This is again an artifact decorrelation of wind speeds between simulations.

To shortly clarify this in the manuscript, the following has been added at the end of section 3: "The slightly positive $\overline{\Delta P}$ in Figure 2c is an artifact of the decorrelation of wind speed between the simulations with and without turbines caused by the presence of the turbines. A longer simulation time would effectively eliminate this artifact and therefore does not affect AEP estimates from year-long simulations."